# DISTILLING CROSS-DOMAIN KNOWLEDGE FOR PERSON RE-ID BY ALIGNING ANY PRETRAINED ENCODER WITH CLIP TEXTUAL FEATURES

## ABSTRACT

Based on the alignment of image-text pairs, CLIP has demonstrated superior performance across various tasks, even in a zero-shot setting. In person ReID, CLIP-based models achieve state-of-the-art results without explicit text descriptions for further fine-tuning. However, previous models are primarily initialized with weights from ImageNet or self-supervised methods, lacking cross-domain knowledge in both image and text areas. This paper introduces a novel approach that aligns a pure image-domain pretrained student model with CLIP textual features, distilling cross-domain knowledge from existing CLIP-ReID into the online student model. To leverage CLIP's textual features for each ID, we address the challenge of mismatched feature dimensions between the teacher and student. A trainable adapter is inserted on the student side to match dimensions and preserve the prior knowledge within the pretrained student. For the student encoder yielding lower or equal-dimensional features compared to the teacher, the adapter is initialized as an identity matrix, while offline PCA is employed on the teacher side for dimensionality reduction. PCA eigenvectors are computed from all training images and applied to existing text features for matching with the student. In cases where the student outputs exceed the teacher's dimensions, the adapter is initialized using eigenvectors computed from the student side to retain knowledge in the pretrained student model. After dimension alignment, text features for each ID are compared with online image features, specifying cross-domain similarities, which are further constrained to mimic the teacher through a KL-divergence loss. Experiments with different pretraining encoder structures demonstrate the effectiveness of this approach, which is also compatible with relation knowledge distillation to enhance performance.

## 1 INTRODUCTION

As a cross-domain model, CLIP Radford et al. (2021) can be directly utilized in many downstream tasks without any fine-tuning. *E.g.*, it is able to classify images by inserting class keywords into a template like "a photo of ...", generating textual features that are compared with visual features from input image to make final predictions without prior training on labeled data. The zero-shot inference capability of CLIP stems from its large number of training data consisting of image-text pairs, and its training scheme which aligns visual and textual representations in a shared embedding space. The cross domain alignment allows it to leverage knowledge from diverse data sources, making it particularly effective for tasks where annotated data may be scarce or absent.

In person re-identification (ReID), CLIP-based methods Li et al. (2023a); Lin et al. (2023); Zhai et al. (2024) that initialize with pretrained weights and fine-tune on downstream ReID data also demonstrate competitive results. Notably, these approaches eliminate the need for explicit textual descriptions by using trainable text prompt tokens to represent each ID, serving as constraints for optimizing the image encoder. However, most of the previous ReID methods Zhou et al. (2019); He et al. (2021); Luo et al. (2021); Chen et al. (2023), rely on the single domain pretraining model derived only from supervised or self-supervised methods in the image domain. As a result, there are no pretrained text encoders available to provide cross-domain descriptions. Such single-domain methods negatively impacts the model's performance. To bridge the gap between single and cross

domain models, our method introduces a novel approach that aligns a image-domain pretrained student model with CLIP's textual features. This alignment provides a comprehensive solution to the absence of pretrained text encoders, enabling the matching textual features of each ID for any single domain image encoder. By effectively leveraging the strengths of both visual and textual domains, we distill cross-domain knowledge from the teacher model, which is the CLIP-ReID Li et al. (2023a), to enhance the student model's performance.

Particularly, to tackle the challenge of mismatched feature dimensions between the teacher and student, a trainable adapter is inserted on the student side to ensure dimensional consistency while preserving the prior knowledge in the pretrained student model. The adapter is a simple linear layer with initialization schemes that vary based on the dimension comparison between the teacher and student. For cases where the student outputs features of lower or equal dimensions compared to the teacher, the adapter is initialized as an identity matrix. Meanwhile, we employ offline PCA on the teacher model for dimensionality reduction, using PCA eigenvectors computed from the training images to align existing text features with the student. When the student outputs exceed the teacher's dimensions, the adapter utilizes eigenvectors derived from the student model as the initial parameters to retain valuable knowledge. After achieving dimension alignment, we compare the text features of each ID with the online image features, specifying cross-domain similarities and constraining them through a KL-divergence loss to emulate the teacher model's performance. To validate the effectiveness of our approach, we conduct experiments on various backbones, including TinyViT Wu et al. (2022), OSNet Zhou et al. (2019), and Solider Chen et al. (2023), which are pretrained using either supervised learning on ImageNet or self-supervised learning on LUPerson Fu et al. (2021). In summary, the contributions of this paper are as follows.

- We propose to align the image-domain pretrained backbone with existing textual features that describe each ID, creating a text-image cross-domain model. We address three scenarios in which the teacher's feature dimension is larger than, equal to, or smaller than that of the student. To ensure dimensional alignment while preserving the knowledge within the student model, we utilize a parametric adapter with tailored initialization schemes for each scenario.

- Our findings indicate that knowledge distillation using a pretrained CLIP-based ReID model as the teacher can significantly enhance the student's performance. Notably, the cross-domain and relational knowledge distillation approaches are compatible in ReID tasks, effectively compensating for the triplet and ID loss during supervised fine-tuning.

- We conduct extensive experiments across various person ReID datasets to demonstrate the effectiveness of our method. Specifically, we achieve state-of-the-art results on both the Market1501 and MSMT17 datasets using the Soldier pretrained backbone.

## 2 RELATED WORKS

**Supervised Person ReID** is a common representation learning task. CNN-based models, particularly those built on ResNet-50 Luo et al. (2019); Dai et al. (2019); Ye et al. (2022) pretrained on ImageNet, have been widely adopted across various ReID datasets. These models are typically optimized using a combination of ID classification loss and metric learning, specifically the triplet loss Hermans et al. (2017), to reduce the distances between features of the same ID in the embedding space, ensuring the model can generalize to unseen IDs during inference. However, CNNs often focus on small, irrelevant regions in the spatial feature maps, limiting the effectiveness of their feature representations. To address this, researchers have introduced attention layers on top of CNNs Chen et al. (2019); Wang et al. (2022a) to expand the receptive field. In addition to global representations, local part features Sun et al. (2018); Wang et al. (2018); Li et al. (2021) and semantic parts Kalayeh et al. (2018); Zhu et al. (2020) have also proven effective in learning more discriminative features. Besides ResNet-50, other light-weight CNN backbones like Zhou et al. (2019); Gu et al. (2023) are also proposed. They have advantages on training and inference speed.

Image transformers Dosovitskiy et al. (2021) have recently gained popularity in ReID tasks. Like CNN-based models, they are also pretrained on ImageNet. Models such as TransReID He et al. (2021), AAFormer Zhu et al. (2021), DCAL Zhu et al. (2022a), DC-Former Li et al. (2023b), and PFD Wang et al. (2022b) have achieved better performance than CNN-based approaches, particularly on large and challenging ReID datasets like MSMT. Furthermore, self-supervised pretraining

methods built on larger datasets, such as LUPerson Fu et al. (2021), including TransReID-SSL Luo et al. (2021), PASS Zhu et al. (2022b), and SOLIDER Chen et al. (2023), have further boosted performance.

All of the above methods utilize models pretrained in a single image domain. CLIP-ReID Li et al. (2023a) is the first work to leverage CLIP's cross-domain pretraining and has achieved state-of-the-art results on MSMT, demonstrating the potential of cross-domain pretraining even for single-domain ReID tasks. Thus, we select CLIP-ReID as the teacher model, aligning its textual features for each ID with a single-domain model, thereby incorporating cross-domain constraints during optimization.

**Knowledge Distillation** Hinton (2015) encourages the online student model to mimic the teacher model. Typically, the teacher has more parameters than the student and remains fixed during training. By applying a distillation loss that compares the outputs of the teacher and student models, the so-called "dark knowledge" from the teacher can be transferred to the student. For classification tasks, this distillation loss is often computed using KL divergence, which measures the difference between the two predicted logit vectors produced by softmax functions. Distillation can also be performed in other forms, such as feature distance Romero et al. (2014), pairwise similarity Tung & Mori (2019); Park et al. (2019), or through contrastive learning Tian et al. (2019). The teacher model can be updated online during training Tarvainen & Valpola (2017). Recently, knowledge distillation has been widely adopted for training lightweight models He et al. (2022); Wu et al. (2022), and even for ViT Touvron et al. (2021). It can also be performed more efficiently through fast distillation methods Shen & Xing (2022). Moreover, online distillation has proven useful for cross-domain pretraining on image-text pairs Dong et al. (2023).

## 3 METHOD

We propose a novel knowledge distillation scheme for supervised learning in person ReID. Our key idea is to leverage a CLIP-based model as the teacher, guiding the optimization of the student model by incorporating a cross-domain constraint to mimic the teacher. The overview of the proposed method is shown in Figure 1. Formally, we denote $\mathcal{T}$ as the teacher model, which consists of an image encoder $E_I$ and text encoder $E_T$ structured as transformers. The image features $f_I = E_I(I) \in \mathbb{R}^d$ output by $E_I$ represent each image and are used in the inference stage for pairwise distance calculations, where $d$ is the feature dimension.

### 3.1 PRELIMINARIES: CLIP-BASED REID MODEL

In the CLIP-based ReID model, the pretrained weights for both $E_I$ and $E_T$ already contain rich prior knowledge derived from a large number of image-text pairs. However, since ReID is a fine-grained task, $E_I$ requires fine-tuning based on the specific downstream data. The optimization of $E_I$ is conducted in two stages. In the first stage, learnable prompt tokens shared for each identity are inserted into a predefined text template $T$ of "a photo of a [X][X][X][X] person" and fed into the text encoder$E_T$, generating textual features $f_T = E_T(T)$ for each ID. During this stage, only the prompt tokens [X] are optimized while all parameters in $E_I$ and $E_T$ remain fixed. The training objective is defined by a multi-positive contrastive loss, as illustrated in Equations 1 and 2..

$$\mathcal{L}_{i2t}(i) = -\frac{1}{|P(i)|} \sum_{p \in P(i)} \log \frac{\exp(\mathrm{sim}(f_I^i \cdot f_T^p))/\tau}{\sum_{j \in A(i)} \exp(\mathrm{sim}(f_I^i \cdot f_T^j))/\tau} \tag{1}$$

$$\mathcal{L}_{t2i}(i) = -\frac{1}{|P(i)|} \sum_{p \in P(i)} \log \frac{\exp(f_T^i \cdot f_I^p)/\tau}{\sum_{j \in A(i)} \exp(f_T^i \cdot f_I^j)/\tau} \tag{2}$$

Here, $f_I^i$ and $f_T^i$ are the $i$th image and text within a training batch, respectively. $P(i)$ is the list of positives sharing the same ID labels as $f_I^i$ or $f_T^i$. $f_I^p$ and $f_T^p$ denote the $p$th positive in this list, while $A(i)$ is the set of all images or texts in the batch, excluding $f_I^i$ or $f_T^i$ itself. In the second training stage, $E_T$ and prompt tokens [X] are kept fixed, and together providing textual features $f_T$ for all $C$ identities. During this stage, only parameters within $E_I$ are optimized. Besides $\mathcal{L}_{i2tce}$ defined in Equation 3, they are trained under a common ID $\mathcal{L}_{id}$ and triplet loss $\mathcal{L}_{tri}$. In 3, $y_i$ is the label of the

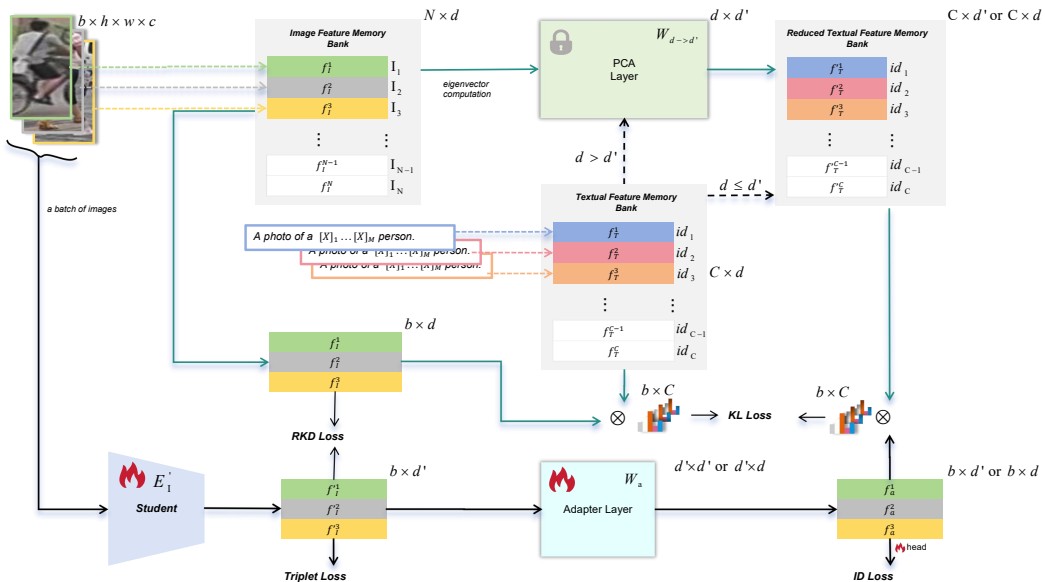

Figure 1: Overview of our proposed method. We use CLIP-ReID Li et al. (2023a) as the teacher model, with memory banks for image and textual features built offline. These stored features provide cross-domain similarity guidance to the student. A trainable linear adapter is inserted into the student model to align its feature dimension $d'$ with the teacher's $d$, initialized in a specific way. Optionally, PCA is applied at the teacher side to reduce the textual feature dimension for better alignment with the student. The KL divergence loss is then calculated, along with relational distillation loss (RKD) and other losses commonly used in ReID tasks, to optimize the student model including the adapter.

$f_I^i$ and $N$ represents the batch size. $\text{sim}(\cdot)$ computes the cross domain cosine similarity of two input vectors.

$$\mathcal{L}_{i2tce}(i) = -\frac{1}{N} \sum_{i=1}^{N} \log \frac{\exp(\text{sim}(f_I^i \cdot f_T^{y_i})/\tau)}{\sum_{j=1}^{C} \exp(\text{sim}(f_I^i \cdot f_T^j)/\tau)} \tag{3}$$

## 3.2 ALIGNING TEXTUAL FEATURES FOR ANY PRETRAINED IMAGE ENCODER

To fully utilize the cross domain description ability of CLIP-based ReID model, we consider to match textual features of all identities with any pretrained image encoder, therefore, converting the single domain image encoder $E_I'$ into the cross domain model including a pair of $E_I'$ and $E_T$. The main challenge is to align the dimension between $f_I' \in \mathbb{R}^{d'}$ in $d'$ dimensional space with $f_T$ in $d$ space. We consider following cases according to comparison results between $d$ and $d'$.

### 3.2.1 DIMENSION REDUCTION AT THE TEACHER SIDE

When the dimensionality of the textual feature $f_T$ exceeds that of the image feature $f_I'$ (i.e., $d > d'$), it is necessary to reduce the dimensions of $f_T$. For this purpose, we employ Principal Component Analysis (PCA). Specifically, we collect all image features from the training data using $E_I$. We then compute the $d'$ eigenvectors corresponding to the largest eigenvalues in the feature space of $f_I$, which collectively form a projection matrix denoted as $W_{d \to d'} \in \mathbb{R}^{d' \times d}$. Since CLIP-ReID keeps the cross domain alignment characteristic of the original CLIP, this matrix is then applied to all textual features $f_T^i$ describing each identity, where $i = 1, 2, \ldots, C$, resulting in the dimension-reduced textual features $f_T^{i'} \in \mathbb{R}^{d'}$. The PCA retains most of the relevant dimensions in the teacher model while minimizing information loss. Note that the dimension reduction at the teacher side is a fixed operation, where $W_{d \to d'}$ is not learnable and remains static. Moreover, $W_{d \to d'}$ is omitted when the student gives the larger dimension which $d \le d'$.

### 3.2.2 Learnable adapter at the student side and its initialization

The online student model $E'_I$ must be fine-tuned for person ReID while simultaneously aligning with the teacher model. To leverage the pretrained weights of $E'_I$ and utilize its prior knowledge, we aim to make minimal modifications without introducing numerous randomly initialized parameters. Therefore, we insert a simple linear learnable layer, parameterized by $W_a$, at the output of $E'_I$ to align $f'_I$ with the teacher model's features. The dimensions of $W_a$ depend on two cases: when $d \geq d'$ or when $d < d'$. In the first case, where the student's feature dimension is greater than or equal to the teacher's ($d \geq d'$), $W_a \in \mathbb{R}^{d' \times d'}$ maintains the feature dimension from $E'_I$. In the second case, where the student's feature dimension is less than the teacher's ($d < d'$), $W_a \in \mathbb{R}^{d \times d'}$ reduces the dimension to $d$.

Furthermore, it is important to initialize $W_a$ appropriately. For $W_a \in \mathbb{R}^{d' \times d'}$, we initialize it as the identity matrix $I$. In other words, at the beginning of training, the features $f'_I$ from the student backbone $E'_I$ are directly passed through the adapter unchanged, ensuring a warm start. In the case where $W_a \in \mathbb{R}^{d \times d'}$, we initialize it using PCA eigenvectors computed from all training images processed by the pretrained $E'_I$. This initialization allows the adapter to capture the most significant variance, preserving essential information and facilitating effective alignment with the feature space of the teacher. In practice, we find that this straightforward linear adapter and its initialization scheme not only facilitate teacher-student alignment but also ensure compatibility of the cross-domain knowledge distillation loss with other loss functions.

### 3.3 Optimization

We now present the details of the optimization process for the student model. Specifically, all parameters in $E'_I$ are trained under two types of losses. The first is the knowledge distillation loss, which ensures that the online student model mimics the output of the teacher model. The second is the traditional ReID loss, which has been proven effective in most ReID models.

### 3.3.1 Knowledge Distillation Loss

The image feature $f_a = W_a f'_I$, produced by the adapter, is used to compute cross-domain similarity with either the original textual feature $f_T$ or the dimension-reduced feature $f'_T$ from the CLIP-based ReID model. While directly applying the $\mathcal{L}_{i2tce}$ loss defined in Equation 3 improves performance, we observe that it yields inferior results compared to cross-domain distillation, which uses the KL divergence loss defined in Equation 4.

$$\mathcal{L}_{kl}(i) = -\frac{1}{N} \sum_{i=1}^{N} \sum_{j=1}^{C} \boldsymbol{q}_i(j) \log \frac{\boldsymbol{p}_i(j)}{\boldsymbol{q}_i(j)} \tag{4}$$

Here, $\boldsymbol{q}_i$ and $\boldsymbol{p}_i$ are cross-domain similarity vectors for the $i$-th image, computed from the teacher and student models, respectively. The $j$-th element in these vectors, such as $\boldsymbol{p}_i(j)$, can be computed as follows, where $f_T^j$ is the textual feature for the $j$-th ID:

$$\boldsymbol{p}_i(j) = \frac{\exp(\mathrm{sim}(f_a^i \cdot f_T^j)/\tau)}{\sum_{k=1}^{C} \exp(\mathrm{sim}(f_a^i \cdot f_T^k)/\tau)} \tag{5}$$

Besides the distillation loss in Equation 4, the pairwise relationships between image features also encapsulate knowledge from the teacher model. Therefore, the relation knowledge distillation loss $\mathcal{L}_{rkd}$ proposed by can also be incorporated. In practice, we find that $\mathcal{L}_{rkd}$ is compatible with $\mathcal{L}_{kl}$ when it is applied to the image feature $f'_I$ before the adapter. In other words, the adapter facilitates the simultaneous use of both distillation losses, enhancing the overall performance of the model.

Since the computation of the knowledge distillation losses, including $\mathcal{L}_{kl}$ and $\mathcal{L}_{rkd}$, involves both the image features $f_I$ and the textual features $f_T$ from the teacher model, as well as their corresponding versions $f'_I$ and $f'_T$ (or the original $f_T$) from the student model, performing all these calculations online would require significant computational resources. To facilitate faster computation for $\mathcal{L}_{kl}$ and $\mathcal{L}_{rkd}$, we adopt strategies from FKD . Specifically, all necessary image and textual features $f_I$ and $f_T$ (or their dimension-reduced counterparts $f'_T$) are pre-computed offline and stored in a memory bank. During training, these features are retrieved from the bank based on the current

| Dataset | Image | Training IDs | Query IDs | Gallery IDs | Cam + View |
|---------|-------|--------------|-----------|-------------|------------|
| MSMT17 | 126,441 | 1,041 | 3,060 | 3,060 | 15 |
| Market-1501 | 32,668 | 751 | 750 | 751 | 6 |
| DukeMTMC-reID | 36,411 | 702 | 702 | 1,110 | 8 |
| Occluded-Duke | 35,489 | 702 | 519 | 1,110 | 8 |

Table 1: Statistics of datasets used in the paper.

online batch. The similarity vector $q$ in 4 and pairwise relation vector in $\mathcal{L}_{rkd}$ is then computed to guide the online version, thus reducing the computational burden on the teacher model during online processing.

### 3.3.2 ReID LOSS

For supervised person ReID, the cross-entropy loss $\mathcal{L}_{id}$ and the triplet loss $\mathcal{L}_{tri}$ are two commonly used loss functions for optimization. These two losses are also compatible with the knowledge distillation losses $\mathcal{L}_{kl}$ and $\mathcal{L}_{rkd}$. Specifically, $\mathcal{L}_{rkd}$ serves as a complement to $\mathcal{L}_{tri}$, while $\mathcal{L}_{kl}$ enhances $\mathcal{L}_{id}$. By integrating these loss functions, we can effectively leverage the strengths of both traditional ReID training and knowledge distillation techniques. In the experiment section, we verify the effectiveness of each loss term.

## 4 EXPERIMENTS

We conduct experiments on four different person ReID datasets, including MSMT17 Wei et al. (2018), Market-1501 Zheng et al. (2015), DukeMTMC-reID Ristani et al. (2016), Occluded-Duke Miao et al. (2019). To evaluate performance, we use cumulative matching characteristics (CMC) at Rank-1 (R1) and mean average precision (mAP). The statistics of each training dataset are listed in Table 1. Note that MSMT17 is the largest and most challenge dataset in person ReID. So we emphasize the performance on this dataset and do ablation study on it.

### 4.1 IMPLEMENTATION DETAILS

We utilize CLIP-ReID Li et al. (2023a) as the teacher model. It provides us image and textual features from its image and text encoder $E_I$ and $E_T$, respectively. Particularly, we choose the ViT-B model without any camera ID as input for side information embedding. We only use the cross domain (post) layer, which is the last layer of CLIP-ReID, and both the image and textual feature $f_I, f_T \in \mathbb{R}^{512}$. Features from middle and previous layer in the teacher model are not considered during the distillation. We follow most of the training set in CLIP-ReID. Images are resized into the resolution of $256 \times 128$. Each mini-batch consists of $B = P \times K$ images, where $P = 16$ represents the number of randomly selected identities, and $K = 4$ indicates the number of samples per identity. For data augmentation, random horizontal flipping, padding, cropping, and erasing are conducted on the student side.

We choose TinyViT (11M) Wu et al. (2022), OSNet Zhou et al. (2019), Solider-Tiny, -Small and -Base models Chen et al. (2023) as the student for optimization. All these models are pretrained within the single image domain. The first two are pretrained on ImageNet-22k and -1k in supervised way. The Solider series are pretrained by self-supervised learning. For TinyViT, image features $f'_I$ lie in lower dimensional space than the teacher, which means $d' = 448 < d$. Hence, the learnable adapter is in size of $448 \times 448$ and initialized by the identity matrix. Offline PCA dimension reduction is performed on textual features from the teacher model. For OSNet, image feature dimension $d' = 512 = d$, and the adapter is $512 \times 512$ and initialized in the same way. We omit PCA at the student side. For Solider series models, all models have larger dimension than the teacher with $d' > d$. The tiny and small models have $d' = 768$ and the adapter is of $768 \times 512$. While the base model has $d' = 1024$, and the adapter is $1024 \times 512$. They are initialized by PCA eigen vectors computed from all training images. For TinyViT, it is trained for 90 epochs with Cosine annealing for scheduling the learning rate. For optimization of OSNet and Solider series, it is trained for 120 epochs. Extra training details are provided at the Appendix.

| Pretraining dataset | Backbones | Methods | MSMT17 | | Market-1501 | | DukeMTMC | | Occ-Duke | |
|---|---|---|---|---|---|---|---|---|---|---|
| | | | mAP | Rank-1 | mAP | Rank-1 | mAP | Rank-1 | mAP | Rank-1 |
| LAINON-5B | | CLIP-ReID | 63.0 | 84.4 | 89.8 | 95.7 | 80.7 | 90.0 | 53.5 | 61.0 |
| ImageNet | ResNet-50 | PCB* | - | - | 81.6 | 93.8 | 69.3 | 83.3 | - | - |
| | | MGN* | - | - | 86.9 | 95.7 | 78.4 | 88.7 | - | - |
| | | ABD-Net* | 60.8 | 82.3 | 88.3 | 95.6 | 78.6 | 89.0 | - | - |
| | | HOReID | - | - | 84.9 | 94.2 | 75.6 | 86.9 | 43.8 | 55.1 |
| | | ISP | - | - | 88.6 | 95.3 | 80.0 | 89.6 | 52.3 | 62.8 |
| | | SAN | 55.7 | 79.2 | 88.0 | **96.1** | 75.5 | 87.9 | - | - |
| | | PAT | - | - | 88.0 | 95.4 | 78.2 | 88.8 | 53.6 | 64.5 |
| | | CAL* | 56.2 | 79.5 | 87.0 | 94.5 | 76.4 | 87.2 | - | - |
| | | CBDB-Net* | - | - | 85.0 | 94.4 | 74.3 | 87.7 | 38.9 | 50.9 |
| | | DRL-Net | 55.3 | 78.4 | 86.9 | 94.7 | 76.6 | 88.1 | 50.8 | 65.0 |
| | | C2F | - | - | 87.7 | 94.8 | 74.9 | 87.4 | - | - |
| | Light CNN | Auto-ReID* | 52.5 | 78.2 | 85.1 | 94.5 | - | - | - | - |
| | | OSNet | 52.9 | 78.7 | 84.9 | 94.8 | 73.5 | 88.6 | 43.4 | 53.0 |
| | | CDNet | 54.7 | 78.9 | 86.0 | 95.1 | 76.8 | 88.6 | - | - |
| | | MSINet | 59.6 | 81.0 | 89.6 | 95.3 | - | - | - | - |
| | | OSNet-KD | 61.2 | 82.5 | 87.7 | 94.8 | 78.4 | 88.3 | 50.0 | 57.1 |
| LAION-5B | | CLIP-ReID | 73.4 | 88.7 | 89.6 | 95.5 | 82.5 | 90.0 | 59.5 | 67.1 |
| ImageNet | ViT-B | AAformer* | 63.2 | 83.6 | 87.7 | 95.4 | 80.0 | 90.1 | 58.2 | 67.0 |
| | | TransReID! | 64.9 | 83.3 | 88.2 | 95.0 | 80.6 | 89.6 | 55.7 | 64.2 |
| | | TransReID!* | 69.4 | 86.2 | 89.5 | 95.2 | 82.6 | 90.7 | - | - |
| | | DiP | 67.5 | 84.6 | 90.3 | 95.7 | 83.8 | 91.2 | 59.1 | 66.4 |
| | | PFD | 65.1 | 82.7 | 89.6 | 95.5 | 82.2 | 90.6 | 60.1 | 67.7 |
| | | DCAL | 64.0 | 83.1 | 87.5 | 94.7 | 80.1 | 89.0 | - | - |
| | | X-ReID | 65.1 | 84.0 | 88.0 | 94.9 | - | - | - | - |
| | | DC-Former! | 69.8 | 86.2 | 90.4 | 96.0 | - | - | - | - |
| | TinyViT | TinyViT | 58.2 | 81.7 | 85.3 | 93.7 | 76.6 | 86.5 | 49.5 | 60.5 |
| | | TinyViT-KD | 68.4 | 85.9 | 89.3 | 95.0 | 81.4 | 90.1 | 56.9 | 63.5 |
| LUPerson | SwinT | SoliderT | 67.4 | 85.9 | 91.6 | 96.1 | 82.1 | 91.2 | 56.7 | 66.6 |
| | | SoliderT-KD | 68.6 | 86.5 | 92.2 | 96.2 | 83.1 | 91.3 | 61.4 | 69.0 |
| | SwinS | SoliderS | 76.9 | 90.8 | 93.3 | 96.6 | 85.7 | 92.8 | 66.5 | 75.2 |
| | | SoliderS-KD | 77.8 | 90.9 | 93.7 | 96.9 | 87.1 | 93.3 | 67.5 | 73.7 |
| | SwinB | SoliderB | 77.1 | 90.7 | 93.9 | 96.9 | 85.8 | 92.6 | 64.6 | 72.5 |
| | | SoliderB-KD | 79.0 | 91.1 | 94.1 | 96.9 | 87.3 | 92.9 | 67.9 | 74.7 |

Table 2: Comparison with state-of-the-art methods on four different person ReID datasets. We categorize these methods by their pretraining datasets and backbone structures. The superscript star* means that the input image is resized to a resolution larger than $256 \times 128$, while the exclamation mark ! indicates the utilization of camera information. Our models, including OSNet-KD, TinyViT-KD and Solider-KD, significantly outperform its non-distillation models, meanwhile, they also achieve competitive performance comparing with other methods.

## 4.2 QUANTITATIVE COMPARISON WITH OTHER METHODS

In Table 2, we present a comparative analysis of our proposed method with the state-of-the-art approaches. Particularly, we categorize all methods based on its pretraining datasets, structures of backbones, and conduct experiments on three different types of backbones including TinyViT, OSNet and Solider series. Since we perform knowledge distillation, light-weight models are intentionally chosen. However, we emphasize that on larger model like SwinB, our method is also effective.

On all of the four datasets, the proposed cross domain distillation method is able to boost the performance compared to the corresponding non-distillation model. Particularly, our OSNet-KD model achieves 61.2 mAP and 82.5 R1 on MSMT17 dataset using light-weight CNN backbones, which is even better than ABD-Net in ResNet-50. This metric also surpasses MSINet Gu et al. (2023), which is a more advanced light-weight model. Our TinyViT-KD model, with its mAP of 68.4 and R1 of 85.9, shows a much better performance than TransReID in the advanced setting which uses camera information and larger ViT-B model denoted by "!". It even approaches the highest metric of "!*", which benefits from much larger number of tokens. For Solider series pretraining, KD versions give higher metrics on all datasets compared with its original ones. Note that due to usage of extra human data in LUPerson Fu et al. (2021), the original Solider models have already achieved competitive performances, but their corresponding KD versions still obtain higher metrics on all datasets. All metrics on backbones of SwinS and SwinB surpass the teacher model, namely the CLIP-ReID, by a large margin.

| Method | Params | FLOPs | Dims |
|--------|--------|-------|------|
| TinyViT | 11M | 1.30 | 448 |
| OSNet | 3.5M | 1.01 | 512 |
| SoliderT | 28M | 2.93 | 768 |
| SoliderS | 50M | 5.66 | 768 |
| SoliderB | 88M | 10.01G | 1024 |
| CLIP-ReID | 86M | 35.75G | 1280 |
| TransReID | 86M | 35.75G | 768 |

Table 3: Key parameters of different models that affect the inference efficiency.

### 4.3 ABLATION STUDIES AND VISUALIZATIONS

Detailed ablation studies are carried out on MSMT17 datasets with TinyViT as the backbone, to show the effectiveness of our cross domain distillation scheme and usage of the inserted adapter. The results are listed in Table 4. Specifically, we first conduct series of experiments without bringing in the learnable adapter. This is mainly for checking the utility of each loss term. It is obvious that training ReID model in normal supervised manner by ID loss $\mathcal{L}_{id}$ and triplet loss $\mathcal{L}_{tri}$ only gives an inferior result with mAP of 58.2 and R1 of 81.7. Incorporating any distillation loss, *e.g.*, $\mathcal{L}_{rkd}$, $\mathcal{L}_{kl}$ or $\mathcal{L}_{ckl}$, will give a boost on the metrics. Here $\mathcal{L}_{kl}$ defined in Equation 4 introduces the cross domain similarity from CLIP-based teacher model. $\mathcal{L}_{rkd}$ proposed by Park et al. (2019) is computed based on the pairwise relation of two image features. We also try knowledge distillation from the parametric image classifier in the last layer of the teacher model, which is $\mathcal{L}_{ckl}$, and it is a pure image domain distillation loss without considering the cross domain similarity like $\mathcal{L}_{kl}$. Although any distillation loss gives the positive effect, $\mathcal{L}_{rkl}$ gives the best performance with mAP of 68.0. Moreover, they are not compatible with each other, and using any two of them leads to a worse metric. Then we insert a linear adapter at the student side to make $\mathcal{L}_{kl}$ to be compatible with $\mathcal{L}_{rkd}$. In this case, these two knowledge distillation losses computed on different image features after and before the learnable adapter, respectively. It gives the best mAP of 68.4. However, adding $\mathcal{L}_{ckl}$ or removing $\mathcal{L}_{kl}$ degrades the metric to 67.9 and 66.3, respectively. Moreover, we are curious about the effect of $\mathcal{L}_{tri}$ and $\mathcal{L}_{id}$, and find that removing any of them slightly decreases the metric to 68.2 and 67.7, respectively. We think these downstream task losses become important when optimizing a better pretrained model like Solider series, and helps the student model beat the teacher. More ablations on OSNet can be found in the Appendix, which also gives a similar results.

To further compare the teacher and student models and understand the effect of knowledge distillation losses, TSNE visualization is performed in Figure 2. Particularly, we randomly select 10 identities and encode their images into the cross domain embedding layer after the adapter. Then TSNE reduces these vectors into 2D space. We compare feature spread in the 2D space and find that the student model highly mimic the behaviour of the teacher model, showing that knowledge distillation losses indeed affect the training process for the student model.

We also perform Gradcam Selvaraju et al. (2017) visualizations and compare the results with CLIP-ReID. Gradcam heat-maps aims to explain the model classification results. Usually, smaller model tends to give a reasonable results. Figure 3 shows the results from CLIP-ReID and TinyViT-KD on the top and bottom rows, respectively. Obviously, the bottom row tends to highlight on the body of person, showing that TinyViT-KD have learned reasonable feature representations. Furthermore, we also show person retrieval results in Figure 4.

Since knowledge distillation primarily aims to train models smaller than the teacher, we list and compare the key parameters of different backbones in Table 3 to highlight the advantages of our smaller models. From this comparison, we observe that TinyViT and OSNet are significantly smaller than other models, with much lower feature dimensions. This is particularly beneficial for ReID inference, as a large number of gallery images typically have pre-computed features, and online inference mainly involves forwarding the query image and calculating a vast number of distances. Thus, a compact feature representation directly improves inference efficiency.

| Adapter Layer | $\mathcal{L}_{tri}$ | $\mathcal{L}_{id}$ | $\mathcal{L}_{rkd}$ | $\mathcal{L}_{kl}$ | $\mathcal{L}_{ckl}$ | mAP | R1 |
|---|---|---|---|---|---|---|---|
| - | ✓ | ✓ | - | - | - | 58.2 | 81.7 |
| - | ✓ | ✓ | ✓ | - | - | 68.0 | 85.7 |
| - | ✓ | ✓ | - | ✓ | - | 65.7 | 85.2 |
| - | ✓ | ✓ | - | - | ✓ | 59.5 | 81.7 |
| - | ✓ | ✓ | ✓ | ✓ | - | 67.5 | 85.3 |
| - | ✓ | ✓ | - | ✓ | ✓ | 67.1 | 85.5 |
| ✓ | ✓ | ✓ | ✓ | ✓ | - | 68.4 | 85.9 |
| ✓ | ✓ | ✓ | ✓ | ✓ | ✓ | 67.9 | 85.5 |
| ✓ | ✓ | ✓ | ✓ | - | - | 66.3 | 85.9 |
| ✓ | ✓ | - | ✓ | ✓ | - | 67.7 | 85.9 |
| ✓ | - | ✓ | ✓ | ✓ | - | 68.2 | 85.8 |
| ✓ | - | - | ✓ | ✓ | ✓ | 67.9 | 85.5 |

Table 4: Ablation study on the adapter at the student side and different combinations schemes of training losses. In all these experiments, we employ TinyViT as the online student model and optimize it on MSMT17 dataset.

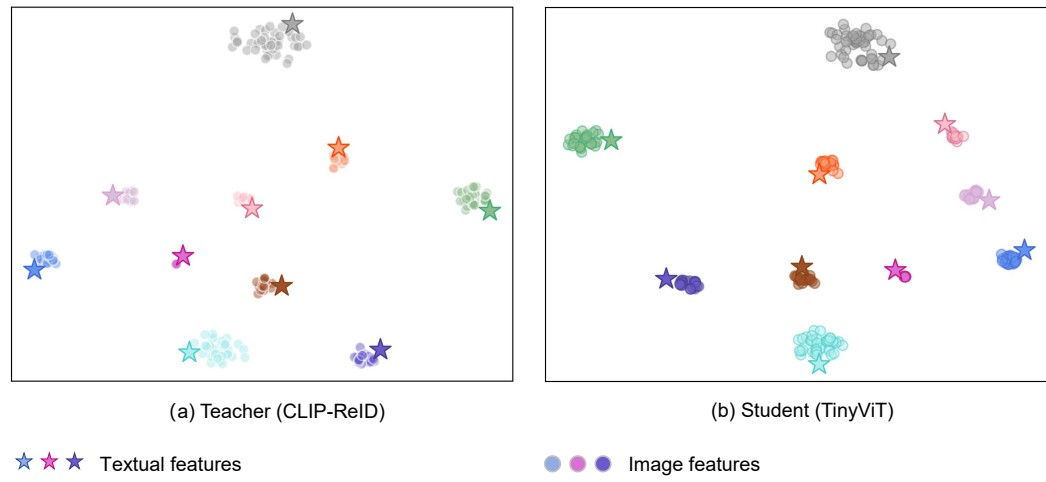

(a) Teacher (CLIP-ReID)          (b) Student (TinyViT)

★ ★ ★  Textual features          ● ● ●  Image features

Figure 2: T-SNE visualizations of the image and textual features from the teacher and student models, respectively. It is obvious that features from the student model have a similar spreading with those in the teacher model.

## 5  CONCLUSION

This paper proposes a cross-domain knowledge distillation method for person ReID. We utilize an optimized CLIP-based ReID model as the teacher to provide cross-domain similarity and pairwise relational guidance to the online student model, which is pretrained solely in the image domain using either supervised or self-supervised methods. An adapter is incorporated on the student side to align its output with the cross-domain embedding of the CLIP-based model. Depending on the dimensional relationship between the student and teacher models, we adopt different strategies for initializing the adapter to better preserve the knowledge within the student model, and optionally reduce textual feature dimension at the teacher side to compute the cross domain similarity for the online model. Constrained by the relational and cross domain KD loss imposed before and after the adapter, our method significantly enhances the performance of different backbones on multiple ReID datasets.

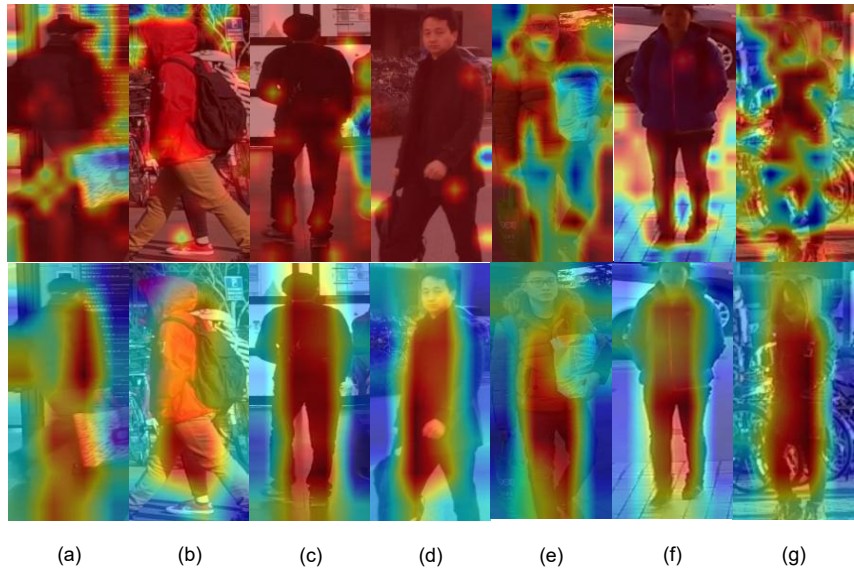

(a)   (b)   (c)   (d)   (e)   (f)   (g)

Figure 3: Gradcam Selvaraju et al. (2017) visualization of our TinyViT-KD model compared with its teacher model CLIP-ReID. On the top, Gradcam results are from the CLIP-ReID. On the bottom, the same images are used for visualization through TinyViT-KD model.

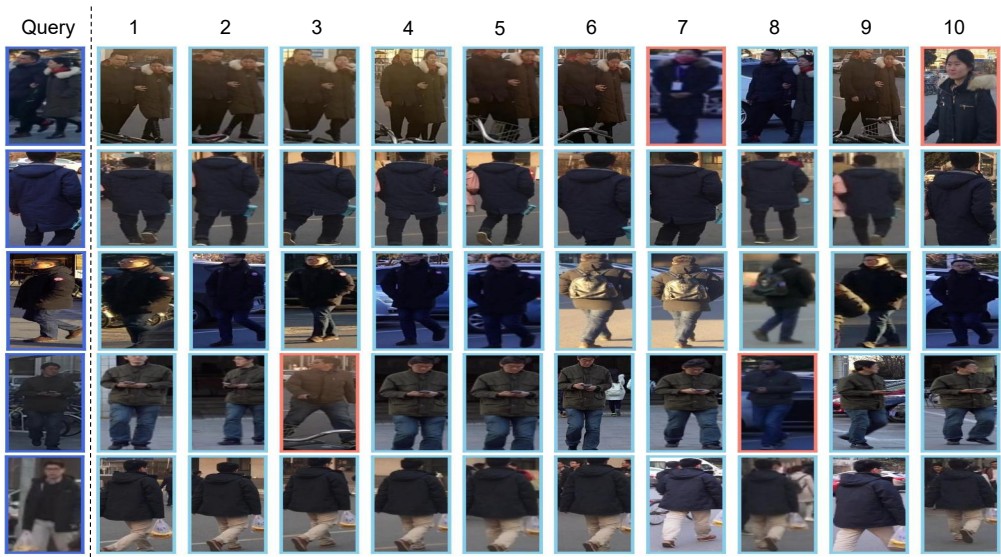

Figure 4: ReID retrieval results. Given a query image, we list its the most similar five images from the gallery set. Images in blue boxes have the same ID with the query, while those in orange boxes are wrong images having different ID with the given query.

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

Table 5: Training details of the different backbones.

| Backbones | Optimizer | Training Epochs | Initial LR | Weight Decay | Warmup Epochs |
|---|---|---|---|---|---|
| TinyViT | AdamW | 90 | $10^{-3}$ | $10^{-2}$ | 5 |
| OSNet | AdamW | 120 | $5 \times 10^{-4}$ | $10^{-4}$ | 20 |
| SoliderT | SGD | 120 | $8 \times 10^{-4}$ | $10^{-4}$ | 20 |
| SoliderS | SGD | 120 | $2 \times 10^{-4}$ | $10^{-4}$ | 20 |
| SoliderB | SGD | 120 | $2 \times 10^{-4}$ | $10^{-4}$ | 20 |

Table 6: Ablation studies on OSnet.

| Adapter Layer | $\mathcal{L}_{tri}$ | $\mathcal{L}_{id}$ | $\mathcal{L}_{rkd}$ | $\mathcal{L}_{kl}$ | mAP | R1 |
|---|---|---|---|---|---|---|
| - | ✓ | ✓ | - | - | 53.2 | 78.0 |
| - | ✓ | ✓ | ✓ | - | 60.4 | 81.5 |
| - | ✓ | ✓ | - | ✓ | 54.2 | 78.6 |
| - | ✓ | ✓ | ✓ | ✓ | 60.5 | 81.9 |
| ✓ | ✓ | ✓ | ✓ | ✓ | 61.2 | 82.5 |

# A APPENDIX

## A.1 TRAINING DETAILS

We provide more training details of different student backbones in Table 5. Basically, for training TinyViT, we follow the hyper-parameters in CLIP-ReID Li et al. (2023a). For training OSNet Zhou et al. (2019) and Solider Chen et al. (2023).

## A.2 MORE ABLATION STUDIES ON OSNET

Here we provide more ablation studies with OSNet backbones. Particularly, we find the setting with learnable adapter and with all loss terms gives the best results, which is the same setting as the detailed ablation study in Table 4.

## A.3 MORE VISUALIZATION RESULTS.

Following figures provide more retrieval results for different query images. For one query image, we show the retrieval results from the teacher model on the top and from the student model on the bottom row.

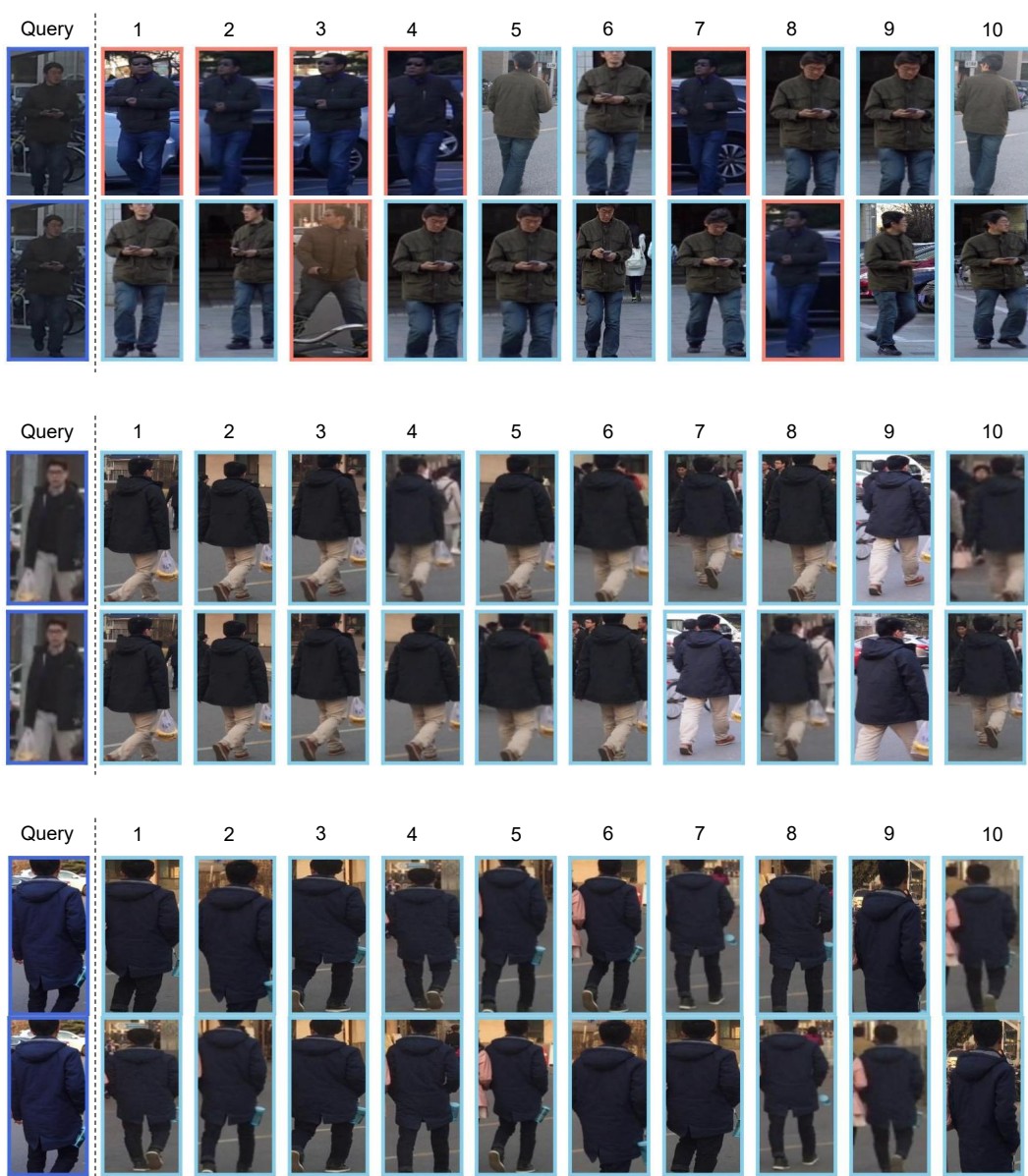

Figure 5: ReID retrieval results comparison between the teacher and student models.

