# OpenReview forum: "Distilling Cross-Domain Knowledge for Person Re-ID by Aligning Any Pretrained Encoder with CLIP Textual Features"
_ICLR.cc/2025/Conference — ICLR 2025 Conference Withdrawn Submission_

### Official Review · Reviewer_teMG · 2024-10-27

**Soundness:** 2
**Presentation:** 2
**Contribution:** 2
**Rating:** 5
**Confidence:** 5

**Summary:**

This paper aims to transfer cross-domain knowledge from existing CLIP-ReID to the student network that trains with only images.

**Strengths:**

The paper is written in a clear and accessible manner.

**Weaknesses:**

1.	The motivation is unclear. Why is it necessary to transfer knowledge from CLIP-ReID to other backbones, and what practical benefits does this bring? When the backbone is smaller than CLIP-ReID, what advantages does knowledge transfer offer? Conversely, when the backbone is larger, how does knowledge transfer contribute? Is this essential?
2.	There is some confusion in the method section:
(1)	Figure 1 is confusing. What does the dotted line in the figure represent? Why does the image feature memory bank become reduced textual feature memory bank after PCA? All in all, the layout and aesthetics of Figure 1 are unsatisfactory.
(2)	Lacking the formula and explanation of the loss term L_{rkd}.
3.	In section 4.3, what are L_{ckl} and L_{rkl}?
4.	Transferring knowledge from CLIP-ReID to TinyViT significantly improves performance. However, this improvement may not solely result from cross-domain knowledge transfer, as the visual encoder in CLIP-ReID is also more powerful than that of TinyViT. It is necessary to demonstrate that textual knowledge plays a positive role.
5.	From Figure 3, Tiny ViT-KD tends to highlight on the body of person. However, why its retrieval performance is inferior to CLIP-ReID?
6.	When the backbone shares a same structure with the visual encoder of CLIP-ReID, what is the effect of transferring textual knowledge or not?

**Questions:**

See Weakness

---

### Official Review · Reviewer_e8Bk · 2024-10-30

**Soundness:** 1
**Presentation:** 2
**Contribution:** 1
**Rating:** 3
**Confidence:** 5

**Summary:**

This paper proposes a cross-domain knowledge distillation method for person ReID. It uses a CLIP-based model as the teacher to guide a pre-trained student model. An adapter aligns the student's output with the teacher's cross-domain embedding. Different initialization strategies are employed based on the model dimensions. The method significantly enhances ReID performance across multiple datasets.

**Strengths:**

1. The structure of this paper is clear.

**Weaknesses:**

The weaknesses of this paper can be summarized as follows:

(1) Limited innovation. The techniques employed in this paper, such as PCA, Adapter, and distillation loss, are widely used across various fields, which limits the novelty of the paper.

(2) Unclear motivation. The paper fails to adequately explain why the proposed method for CLIP-ReID distillation can benefit the Re-ID task. A clearer rationale and explanation are needed to justify the choice of the method.

(3) Poor writing quality. The paper contains bad writing expressions, such as in Page 1, lines 46-48 and so on. Additionally, the images and tables in this paper are not well-presented, and there are incorrect citations throughout the paper.

(4) Inadequate experimentation. The comparative experiments in Table 2 lack references to the latest articles, which makes it difficult to assess the relevance and competitiveness of the results. Furthermore, Table 4 requires in-depth explanations regarding the  effectiveness of each component. Lastly, the existence of Table1 and Table 3,  seems unnecessary and does not contribute significantly to the understanding of the paper's findings.

**Questions:**

(1) Please further elaborate on the motivation behind the designed distillation method with CLIP-ReID

(2) Please explain the reasons and significance for choosing TinyViT (11M), OSNet Zhou Solider-Tiny, -Small, and -Base models as the student models.

---

### Official Review · Reviewer_BvMH · 2024-11-02

**Soundness:** 3
**Presentation:** 3
**Contribution:** 2
**Rating:** 5
**Confidence:** 5

**Summary:**

This paper focuses on the person ReID task and conducts research to address the issue that traditional methods are limited to single-domain pre-trained models and lack cross-domain knowledge. It proposes an innovative method that aligns the image-domain pre-trained student model with the textual features of CLIP to achieve cross-domain knowledge distillation and improve model performance. In response to the challenge of mismatched feature dimensions between the teacher model and student model, this paper develops trainable adapters and diverse initialization strategies to better preserve the knowledge within the student model and optionally reduce the textual feature dimension at the teacher side. Excellent results are achieved on some person ReID datasets, providing new ideas and methods for the research of person ReID.

**Strengths:**

1. This paper innovatively proposes a cross-domain knowledge distillation method based on CLIP-ReID for person re-identification, solving the limitations of traditional single-domain pre-trained models. The idea of applying the CLIP-ReID model to guide the training of lightweight student models and the adapter strategy designed to address the feature dimension mismatch problem is still relatively new attempts in the field of person re-identification(ReID), providing new perspectives for subsequent research.

2. The experimental design is comprehensive, verified on multiple person re-identification datasets with various backbones. the roles of multiple loss functions are analyzed in detail, and the impact of each component on the model performance is deeply explored through ablation experiments.

3. The paper has a clear structure, coherent logic, and accurate language expression. The explanations of complex concepts and technologies are easy to understand, enabling readers to easily understand the research content and method.

**Weaknesses:**

1. There is an inconsistency in symbol definitions in the paper. For example, the symbol representations of the feature dimensions of the student model and the teacher model on lines 221-223 of page 5 are contradictory to those on line 184 of page 3. It is recommended to unify the symbol definitions.

2. The explanation of the experimental results is not deep and comprehensive enough. Although the paper shows the performance improvement of the proposed method on multiple datasets and backbones, the underlying reasons for these improvements are not fully explained. In the comparison experiment part, only the comparison results with other methods are listed, but the roots of the advantages and disadvantages of the proposed method compared with other advanced methods are not deeply explored. It’s recommended to add more in-depth analysis in the comparison experiment to clarify the improvement of the proposed method and thus improve the academic value of the paper.

3. The main work focuses on performing knowledge distillation based on CLIP-ReID and aligning the image domain features of the text domain and the image encoder, essentially adding an operation of aligning text and image features. Compared with the numerous existing studies in this field, this improvement is relatively routine and does not propose an innovative network architecture or method, lacking core innovation points. It is recommended to expand research ideas and explore how to combine this method with other technologies to solve more practical problems and enhance the application potential of the method in practice.

4. The typesetting of this paper seems unreasonable, and the content is not rich enough.

**Questions:**

1. Although attention is paid to the cross-domain feature dimension mismatch problem, are there limitations in the handling of different dimension relationships?

2. Why are there so many '?' marks in the Figure1 of the paper?

---

### Official Review · Reviewer_ruLX · 2024-11-02

**Soundness:** 2
**Presentation:** 2
**Contribution:** 2
**Rating:** 3
**Confidence:** 3

**Summary:**

The paper introduces a novel approach in the field of person re-identification (ReID). The authors propose a method to distill cross-domain knowledge from a CLIP-ReID model into a student model that is pretrained solely in the image domain. The key innovation is aligning the pretrained student model with CLIP's textual features, which provides a comprehensive solution to the absence of pretrained text encoders. This alignment is achieved by inserting a trainable adapter layer in the student model to match feature dimensions between the teacher (CLIP-ReID) and student models. The approach leverages CLIP's textual features for each identity and employs dimensionality reduction techniques like PCA when necessary. The student model is optimized using a KL-divergence loss to mimic the teacher model's performance, along with traditional ReID losses like ID loss and triplet loss. The effectiveness of the approach is demonstrated through extensive experiments on various datasets.

**Strengths:**

1. The paper addresses the challenge of mismatched feature dimensions between teacher and student models by using a trainable adapter and PCA, providing a flexible solution for different model architectures.
2. The proposed method is shown to be effective across various backbones, including TinyViT, OSNet, and Solider, demonstrating its versatility.

**Weaknesses:**

1. I am confused about the motivation of this paper. On the one hand, it is precisely because of the limitations of the ReID model based on unimodal pre-training in terms of semantically rich features that CLIP-ReID proposes to use the multimodal pre-training model CLIP to make up for its limitations. Why does this paper want to distill the knowledge of CLIP-ReID to the ReID model based on unimodal pre-training? On the other hand, since there is already a CLIP-ReID model, why not use it directly, but instead go to the trouble of distilling this knowledge to the ReID model based on unimodal pre-training.
2. Line 137 of Section 3 mentions “The image features $f_I$ output by $E_I$ represent each image and are used in the inference stage for pairwise distance calculations, where d is the feature dimension.”. So the image encoder of the teacher network is also used for inference? Please elaborate on the inference process.
3. I am confused about the process in Section 3.2. When d>d', PCA is applied first and then the adapter is performed. Otherwise, only the adapter is performed. Is my understanding correct?
4. Section 3.3.1, line 260, mentions "This is a dog". There seems to be a missing reference here.
5. Is the loss $L_rkd$ applied to the image features before the adapter is applied? I think it is necessary to show the calculation formula of this loss.
6. What is FKD mentioned in line 268 of section 3.3.1? Is there a missing reference here?

**Questions:**

Please see Weaknesses

---

### Note · Authors · 2024-11-14

I have read and agree with the venue's withdrawal policy on behalf of myself and my co-authors.